# Validation of artificial intelligence-based digital microscopy for automated detection of *Schistosoma haematobium* eggs in urine in Gabon

Brice Meulah[1,2]*, Prosper Oyibo[3], Pytsje T. Hoekstra[1], Paul Alvyn Nguema Moure[2,4], Moustapha Nzamba Maloum[2], Romeo Aime Laclong-Lontchi[2], Yabo Josiane Honkpehedji[1,2,5], Michel Bengtson[1], Cornelis Hokke[1], Paul L. A. M. Corstjens[6], Temitope Agbana[3], Jan Carel Diehl[7], Ayola Akim Adegnika[1,2,4,5,8], Lisette van Lieshout[1]

1 Leiden University Center for Infectious Diseases (LUCID), Leiden University Medical Center, Leiden, The Netherlands, 2 Centre de Recherches Médicales des Lambaréné, CERMEL, Lambaréné, Gabon, 3 Mechanical, Maritime and Material Engineering, Delft University of Technology, Delft, The Netherlands, 4 Ecole doctorale régionale d'Afrique centrale en infectiologie tropicale de Franceville, Gabon, 5 Fondation pour la Recherche Scientifique, Cotonou, Benin, 6 Department of Cell and Chemical Biology, Leiden University Medical Center, Leiden, The Netherlands, 7 Industrial Design Engineering, Delft University of Technology, Delft, The Netherlands, 8 Institut fur Tropenmedizin, Universitat Tubingen, Tubingen, Germany

* b.meulah_tcheubousou@lumc.nl

## Abstract

### Introduction

Schistosomiasis is a significant public health concern, especially in Sub-Saharan Africa. Conventional microscopy is the standard diagnostic method in resource-limited settings, but with limitations, such as the need for expert microscopists. An automated digital microscope with artificial intelligence (Schistoscope), offers a potential solution. This field study aimed to validate the diagnostic performance of the Schistoscope for detecting and quantifying *Schistosoma haematobium* eggs in urine compared to conventional microscopy and to a composite reference standard (CRS) consisting of real-time PCR and the up-converting particle (UCP) lateral flow (LF) test for the detection of schistosome circulating anodic antigen (CAA).

### Methods

Based on a non-inferiority concept, the Schistoscope was evaluated in two parts: study A, consisting of 339 freshly collected urine samples and study B, consisting of 798 fresh urine samples that were also banked as slides for analysis with the Schistoscope. In both studies, the Schistoscope, conventional microscopy, real-time PCR and UCP-LF CAA were performed and samples with all the diagnostic test results were included in the analysis. All diagnostic procedures were performed in a laboratory located in a rural area of Gabon, endemic for *S. haematobium*.

**Data Availability Statement:** All relevant data are within the paper and its Supporting Information files.

**Funding:** This work was funded by NWO-WOTRO Science for Global Development program, grant no. W 07.30318.009 (INSPiRED—INclusive diagnoStics for Poverty REIated parasitic Diseases in Nigeria and Gabon) to LvL. The funders had no role in study design, data collection and analysis, decision to publish, or preparation of the manuscript.

**Competing interests:** The authors have declared that no competing interests exist.

## Results

In study A and B, the Schistoscope demonstrated a sensitivity of 83.1% and 96.3% compared to conventional microscopy, and 62.9% and 78.0% compared to the CRS. The sensitivity of conventional microscopy in study A and B compared to the CRS was 61.9% and 75.2%, respectively, comparable to the Schistoscope. The specificity of the Schistoscope in study A (78.8%) was significantly lower than that of conventional microscopy (96.4%) based on the CRS but comparable in study B (90.9% and 98.0%, respectively).

## Conclusion

Overall, the performance of the Schistoscope was non-inferior to conventional microscopy with a comparable sensitivity, although the specificity varied. The Schistoscope shows promising diagnostic accuracy, particularly for samples with moderate to higher infection intensities as well as for banked sample slides, highlighting the potential for retrospective analysis in resource-limited settings.

## Trial registration

NCT04505046 ClinicalTrials.gov.

## Author summary

Assessment of schistosomiasis control programs is a crucial step to understanding the success rate of these control programs. The Schistoscope: an AI-powered automated digital microscope could overcome the limitations of conventional microscopy in endemic resource limited settings as well as in settings lacking microscopy experts. In this study, we carried out an extensive validation of the Schistoscope's diagnostic performance for diagnosis of urogenital schistosomiasis compared to conventional microscopy as well as more accurate diagnostic tests such as real-time PCR and the up-converting particle (UCP) lateral flow (LF) test for the detection of circulating anodic antigen (CAA) on freshly collected urines. We also assessed the performance of the Schistoscope for the diagnosis of schistosomiasis on banked sample slides, using a simple and sustainable storage method, for approximately two years. Having a tool that can prospectively and retrospectively analyse samples in an easy and sustainable way could facilitate schistosomiasis control programs in settings with little or no access to microscopists. Overall, we found the Schistoscope to be as good as conventional microscopy for the diagnosis of schistosomiasis, and given its downstream advantages of digital health, it would serve as a valuable diagnostic/screening tool in resource limited endemic settings.

## 1. Introduction

Schistosomiasis is a tropical parasitic disease of significant public health concern, with an estimated 700 million individuals at risk of infection in areas known for transmission. Out of approximately 250 million people requiring preventive chemotherapy worldwide, Sub-Saharan Africa, including the centrally located country of Gabon, bears the highest proportion [1–3]. In order to control the disease morbidity and work towards its elimination as a public

health problem, the World Health Organization (WHO) recommends annual preventive chemotherapy using a single dose of praziquantel for all individuals aged two years and above in communities where the prevalence of schistosomiasis is 10% or higher [4]. For communities with a prevalence below 10%, an optional test-and-treat strategy is recommended [4]. In both cases, reliable diagnostic tools are essential to support the monitoring and evaluation of these control strategies [5,6].

Conventional microscopy is the standard diagnostic procedure for schistosomiasis. However, the need for expert microscopists limits its application in resource-limited settings. Real-time polymerase chain reaction (PCR) for amplification and detection of schistosome-specific nucleic acid sequences, as well as a lateral flow test (LF) for the detection of schistosome-specific circulating anodic antigen (CAA), offer higher sensitivity and specificity than conventional microscopy [7,8]. Nevertheless, the requirement for specialized skills and advanced infrastructure currently limits their application in resource-limited settings.

Alternatively, automated digital microscopes have shown promising results in the diagnosis of schistosomiasis by detecting parasite eggs in stool or urine [9–12]. The application of artificial intelligence (AI) algorithms in the diagnosis and surveillance of infectious diseases has received significant attention [13–15]. Automated digital microscopes are designed to capture images of samples with simultaneous analysis by an AI algorithm trained to detect parasite components. Such innovative tools are relatively easy to use and can be customised for rural endemic settings. These tools also have propitious downstream applications including digital health [11,16–18]. In particular for the detection of *S. haematobium* eggs in urine, multiple studies have validated the diagnostic accuracy of AI-based digital microscopes, demonstrating sensitivities ranging from 32% to 91% compared to conventional microscopy, as summarised in a recent review [11].

The Schistoscope is an automated digital microscope with an integrated AI algorithm to detect *S. haematobium* eggs in urine samples. It was developed for use at point-of-need and is relatively easy to operate requiring minimal training [19,20]. The Schistoscope was first assessed in Nigeria for diagnosing urogenital schistosomiasis, revealing a high sensitivity but a rather low specificity compared to conventional microscopy [12]. Based on these results, the AI model was re-designed, retrained and embedded onboard the Schistoscope, and then validated using a set of field sample images, yielding better sensitivity and specificity [21]. A limitation of the previous studies has been the small size of validation sample dataset and the lack of an accurate reference standard. To perform more in-depth validation of the diagnostic accuracy of the Schistoscope in detecting *S. haematobium* eggs, urine samples were collected and analysed in a laboratory setting in Lambaréné, Gabon. The diagnostic performance of the Schistoscope was compared to conventional microscopy as well as to a composite reference standard (CRS), consisting of real-time PCR and UCP-LF CAA.

## 2. Methods

### 2.1. Ethics statement

Ethical approval for the study was obtained from the Comité d'Éthique Institutionnel (CEI) du Centre de Recherches Médicales de Lambaréné in Lambaréné, Gabon (reference no. CEI--CERMEL 005/2020). Prior to sample collection, written consent was obtained from adults and from parents or legal guardians of children and teenagers who wished to participate, indicated by their signatures. To ensure confidentiality and anonymity of the results, unique codes were assigned to the samples. Participants with detectable *S. haematobium* eggs/10 mL of urine based on microscopy were treated with praziquantel (40 mg/Kg of body weight) following local guidelines. The study was registered at ClinicalTrials.gov (NCT04505046).

## 2.2. Study design

The validation study was conducted in Lambaréné and surrounding areas, located in the Moyen-Ogooué province in Gabon, a region known to be endemic for *S. haematobium* with a prevalence of about 30% [22]. It was carried out in two parts: study A and study B (Fig 1). Study A was an independent cross-sectional study focusing on school-age children and adults from whom urine samples were collected and analysed by the Schistoscope, conventional microscopy, real-time PCR and UCP-LF CAA (see details below). Study B was partly embedded in several ongoing studies at Centre de Recherches Médicales de Lambaréné (CERMEL) in Gabon, where urine samples were collected from different populations (school-age children, adults and pregnant women) and analysed with a range of diagnostic methods including conventional microscopy, real-time PCR and UCP-LF CAA (see details below). Microscopy slides were subsequently biobanked at 4˚C for retrospective analysis with the Schistoscope (~2 years later). All diagnostic procedures were conducted at CERMEL.

## 2.3. Sample size calculations

The Schistoscope was assumed to have a sensitivity and specificity non-inferior to conventional microscopy, which were realistically assumed to be 80% and 90%, respectively based on field expert estimates using real-time PCR and UCP-LF CAA. The sample size for both study A and B were determined based on a 30% prevalence of schistosomiasis in Lambaréné and its surrounding areas using a two-sample matched paired design, resulting in a required sample size of 350 urine samples [23]. A power of 80% and a 5% degree of error was considered for the calculations.

## 2.4. Sample collection and processing

Collection of urine samples in study A was carried out starting in 2023 while the urine sample biobanking (study B) was initiated in 2020. Study participants were provided with sterile containers labelled with unique identifiers and instructed to provide urine samples between 11 am and 2 pm. The samples were transported to CERMEL within 2 hours of collection for analysis. Microscopy slides were prepared by pressing 10 mL of homogenised urine through a 25mm membrane (pore size 30 μm; Whatmann International Ltd) with the use of a syringe and a filter holder and transferred onto a glass slide. For study A, the slides were examined on the same day using conventional microscopy and the Schistoscope. For study B, the slides were examined using conventional microscopy and stored at 4˚C for about 2 years awaiting analysis with the Schistoscope. For both studies, 1 mL of urine from each sample was used for UCP-LF CAA analysis and 10 mL of homogenised urine was centrifuged and the resulting 1mL pellet was used for DNA extraction and amplification before biobanking the sample slides for retrospective analysis with the Schistoscope.

## 2.5. Diagnostic methods

**2.5.1. The Schistoscope.** Five Schistoscopes were used in this study (Fig 2A and S1 Video). Analysis was done following the standard operating procedure (S1 Manual) of the Schistoscope. Briefly, the slide was placed on the slide holder of the Schistoscope such that its microscope objective aligned with the filter membrane of the slide. The device's autofocus algorithm positioned the microscope objective in the optimal focal plane. High resolution images of the sample were registered and analysed simultaneously by the integrated AI algorithm. The number of detected eggs (expressed in eggs/10~ml of urine) is displayed on a pop-up result window which also indicated the end of the sample analysis. Detected eggs are

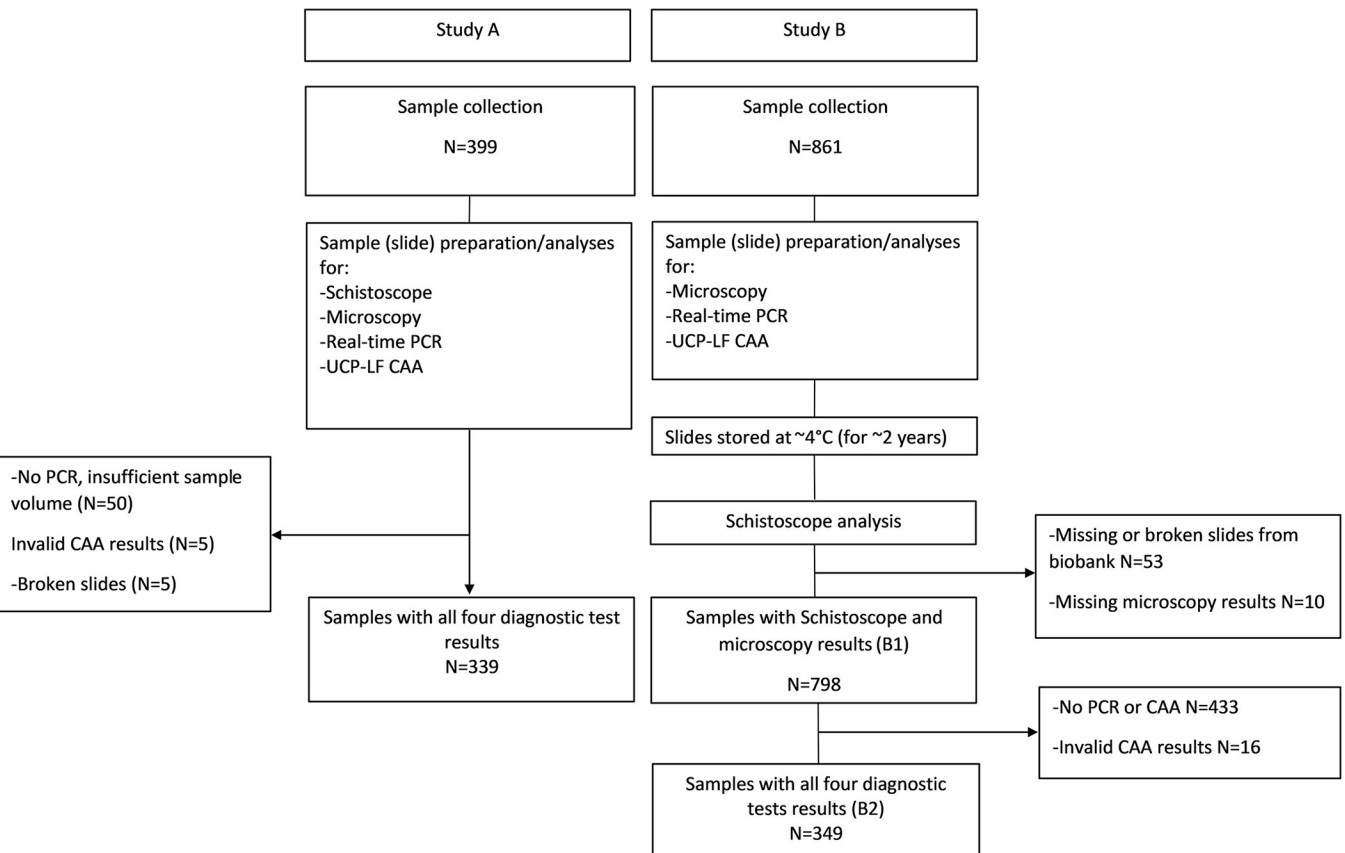

**Fig 1. Comprehensive flow chart detailing the methodical sequence of urine sample collection, processing by the Schistoscope, conventional microscopy, real-time PCR and UCP-LF CAA and data analysis.**

marked as shown in Fig 2B and 2C. The results from the Schistoscope were exported as an Excel-compatible file.

**2.5.2. Conventional microscopy.** Slides from both studies were analysed immediately after urine filtration under 10x objective of a Leica microscope (model: DM1000 LED, Microsystems CMS GmbH Ernt-Leitz-Str.17-37 Wetzlar, Germany). Each slide was examined by two independent microscopists and the mean egg count was calculated. In case of a >20% discrepancy in egg count, an additional reading by a third independent microscopist was required

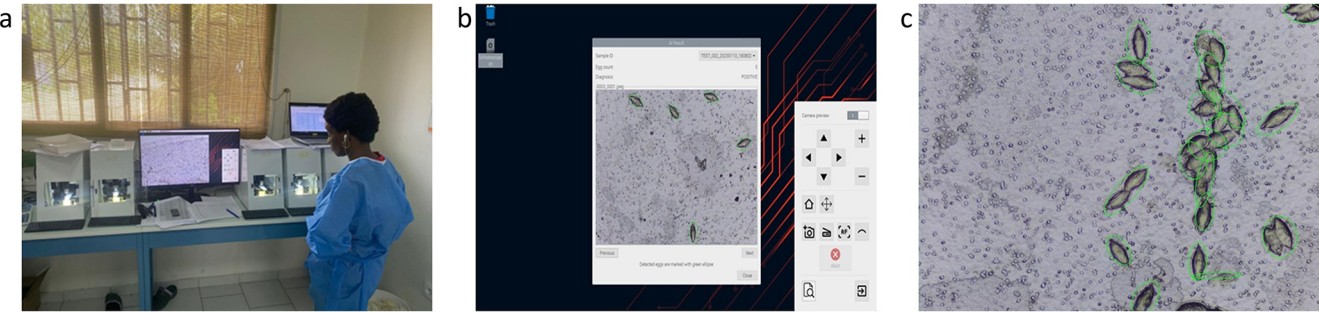

**Fig 2.** (a) Five Schistoscopes connected to a single display and in use for slide analysis by a laboratory technician. (b) Schistoscope display of result window after slide analysis is completed. (c) Schistoscope image showing some of the overlapping eggs counted as a single egg by the AI algorithm.

and the final egg count was determined by calculating the mean of the two closest egg counts obtained from the three readings. All egg counts were expressed as eggs/10 mL of urine. In addition, the storage conditions (4°C) and quality of the biobanked slides were monitored using conventional microscopy once every four months during the storage period. This was done by monitoring daily temperature of the fridge as well as by determining the egg counts of three known slides. Additionally, during the Schistoscope analysis the integrity of the biobank was quality controlled by re-examining a random selection of 10% of the slides by conventional microscopy and comparing the results to the outcomes before storage.

**2.5.3. Nucleic acid extraction and real-time PCR.** Genomic DNA extraction was carried out using the QIAamp Mini kit (cat: 51306; Qiagen) according to the manufacturer's instructions. Briefly, 195μL of each centrifuged urine pellet was mixed with 5μL of internal control DNA commercially available as a DNA Extraction Control (DEC) 670 kit (Cat: BIO-35028; Bioline). The DEC 670 kit is supplied as a vial of internal control DNA sequence (with no known homology to sequences of any organism) and a vial of control mix containing primers and probes complementary to the internal control DNA sequence. The final mixture was then processed as previously described [8].

Real-time PCR was performed as previously described [8, 24] using a set of primers (Ssp48F and Ssp124R) and probe (Ssp78T) complementary to the 77-bp internal transcribed spacer-2 (ITS2) sequence, with minor modifications on the internal control (see above) as well as on the reaction mixture and conditions used (see below).

Amplification reactions were performed in a 15μL reaction mixture containing 1x No-ROX master mix (Cat: BIO-86005; Bioline), 4.5pmol of each *Schistosoma*-specific primer, 1.5pmol Schistosoma-specific probe, 0.4μL of control mix, 1.2μL of nuclease free water and 2.5μL DNA extract. The PCR runs consisted of an initial step of 5 min at 95°C followed by 40 successive cycles of 10 sec at 95°C and 60 sec at 60°C. The reaction was run on a Light cycler 480 II real-time PCR system (Roche Diagnostics). *Schistosoma* DNA detection was expressed in threshold (Ct) cycles. For every run, a non-template control and a positive control (*S. haematobium* DNA, Ct-value 23–25) was included. A test was considered positive when the threshold was attained within 40 PCR cycles (Ct-value $\leq$ 40). Each sample was run in duplicate and was considered positive when at least one of the duplicates was positive. Amplification of the internal control at the expected Ct-value showed success of nucleic acid extraction and no evidence of PCR inhibitors.

**2.5.4. UCP-LF CAA.** Urine CAA concentration was determined by the UCP-LF CAA assay using the UCAA*hT*417 format as previously described [7]. Briefly, 500μL of each urine sample was mixed with 100μL of 12% trichloroacetic acid, incubated and centrifuged. The clear supernatant obtained was concentrated to 20μL using an Amicon Ultra-4 concentration column (Millipore, Merck Chemicals B.V., Amsterdam, The Netherlands) and subsequently mixed with 50μL run buffer and added to 50 μL UCP solution. The resulting mixture was then used for the lateral flow assay. A set of CAA standards was used to validate the cut-off (2 pg/ml) as well as to reliably quantify the amount of CAA per sample up to 1000 pg/ml [7].

## 2.6. Statistical analyses

In study A, only samples with all four diagnostic test results available were included in the final analysis. For study B, samples with both the Schistoscope and conventional microscopy test results only were first analysed (B1). Additionally, a subset of samples (B2) which had outcomes of all four diagnostic tests was analysed separately (Fig 1). The percentage positive samples for a *Schistosoma* infection was determined for each diagnostic test. The sensitivity and specificity of the Schistoscope were assessed using conventional microscopy as the reference

(study A, B1 and B2). Sensitivity and specificity of the Schistoscope and conventional microscopy were further evaluated using a combination of real-time PCR and/or UCP-LF CAA as a CRS (study A and B2). A sample was deemed positive by the CRS if it showed the presence of *Schistosoma* spp DNA and/or CAA. Conversely, a sample was considered negative if both diagnostic tests showed a negative outcome. A ≤10% difference in sensitivity and specificity between the Schistoscope and conventional microscopy based on the CRS was considered non-inferior. To determine the performance of the Schistoscope at different infection intensities, egg counts based on conventional microscopy were categorised into very low intensity infection (1–9 eggs/10 mL), low-intensity infection (10–49 eggs/10 mL) and high-intensity infection (≥50 eggs/10 mL) [25,26]. Cohen's Kappa (k) statistics was computed to assess the qualitative agreement between the Schistoscope and conventional microscopy, and the CRS. Spearman's correlation (r) was used to assess the strength of association between the Schistoscope and conventional microscopy, real-time PCR and UCP-LF CAA. Bland-Altman analysis was further used to assess the quantitative agreement between the Schistoscope and conventional microscopy. Wilcoxon sign rank test was used to compare the microscopy egg count of the randomly selected banked slides before and after storage. Statistical analysis was performed using IBM Statistical Package for Social Sciences version 25 (SPSS Inc., Chicago, United States of America) and GraphPad Prism version 9.0.1 for Windows (GraphPad Software, San Diego, California USA, www.graphpad.com).

## 3. Results

### 3.1. Study A: Diagnostic performance of the Schistoscope on freshly prepared samples

A total of 339 samples had outcomes available for all four diagnostic tests and were included in the analysis. Table 1 shows the proportion of positive results per diagnostic test. Real-time PCR found the highest proportion of positives (51.0%) followed by the UCP-LF CAA assay (46.6%). The proportion of positives detected by the Schistoscope (46.0%) was higher than that of conventional microscopy (38.3%). The median egg count of the Schistoscope (17 eggs/10ml) was lower than that of microscopy (31 eggs/10ml). The proportion of positives with egg count ≥50 eggs/10 mL by the Schistoscope and microscopy were comparable, 47 (30.1%) and 49 (37.7%), respectively (S1A Fig).

Qualitatively, a moderate agreement between the Schistoscope and conventional microscopy was observed (K = 0.579, *P*<0.001). However, the agreement was only fair when compared to the CRS (K = 0.396, *P*<0.001) whereas a moderate agreement was observed between

**Table 1. Diagnostic outcomes of the Schistoscope in comparison to conventional microscopy, real-time PCR and UCP-LF CAA in study A and B.**

| Diagnostic test | Study A (N = 339) | | | | Study B1 (N = 798) | | Study B2 (N = 349) | | | |
|---|---|---|---|---|---|---|---|---|---|---|
| | Schistoscope | Microscopy | Real-time PCR | UCP-LF CAA | Schistoscope | Microscopy | Schistoscope | Microscopy | Real-time PCR | UCP-LF CAA |
| **Positive (%)** | 156 (46.0%) | 130 (38.3%) | 173 (51.0%) | 158 (46.6%) | 374 (46.9%) | 307 (38.5%) | 204 (58.5%) | 190 (54.4%) | 217 (62.2%) | 225 (64.5%) |
| **Range** | 1–1623 eggs/10mL | 1–2516 eggs/10mL | 20.2–37.0 Ct | 2.6–1000.0 pg/mL | 1–2879 eggs/10mL | 1–9350 eggs/10 mL | 1–1943 eggs/10mL | 1–9350 eggs/10mL | 19.1–38.7 Ct | 2.1–1000.0 pg/mL |
| **Median of the positives** | 17 eggs/10mL | 31 eggs/10mL | 29.0 Ct | 65.0 pg/mL | 17 eggs/10mL | 105 eggs/10mL | 32 eggs/10mL | 209 eggs/10mL | 26.6 Ct | 189.6 pg/mL |
| **Mean of the positives** | 78 eggs/10mL | 119 eggs/10mL | 29.7 Ct | 134.0 pg/mL | 136 eggs/10mL | 464 eggs/10mL | 160 eggs/10mL | 565 eggs/10mL | 27.8 Ct | 310.5 pg/mL |

conventional microscopy and the CRS (K = 0.537, $P<0.001$) (Table 2). The sensitivity and specificity of the Schistoscope were 83.1% and 77.0%, respectively, when conventional microscopy was used as reference. In addition, when the Schistoscope and conventional microscopy were evaluated using the CRS, the sensitivity of the Schistoscope was 62.9% comparable to that of conventional microscopy, 61.9%. However, the specificity of the Schistoscope was significantly lower compared to the specificity of conventional microscopy. All samples with an egg count of ≥50 eggs/10mL defined by conventional microscopy were detected by the Schistoscope (S1A Fig). Of the microscopy positive samples with 1–9 eggs/10mL and 10–49 eggs/10mL, the Schistoscope detected 52.6% and 90.7% respectively. Conversely, the Schistoscope found 48 additional cases (of which 40 had <50 eggs/10mL) which were not detected by conventional microscopy. Of these additional cases, 35.4% and 27.1% were confirmed by real-time PCR and the UCP-LF CAA assay, respectively.

**Table 2. Diagnostic performance and pairwise level of agreement by Cohen's Kappa statistics between the Schistoscope and conventional microscopy and the composite reference for the detection of *S. haematobium* infection in study A and B.**

| Sample set | Diagnostic test | Reference test | | Diagnostic test Sensitivity % (95% CI) | Diagnostic test Specificity % (95% CI) | Kappa | P value | Interpretation* |
|---|---|---|---|---|---|---|---|---|
| Study A (N = 339) | | **Microscopy** | | | | | | |
| | **Schistoscope** | Positive | Negative | 83.1 (75.5–89.1) | 77.0 (70.7–82.5) | 0.579 | <0.001 | Moderate |
| | Positive | 108 | 48 | | | | | |
| | Negative | 22 | 161 | | | | | |
| | | **Composite reference** | | | | | | |
| | **Schistoscope** | Positive | Negative | 62.9 (55.8–69.6) | 78.8 (71.0–85.3) | 0.396 | <0.001 | Fair |
| | Positive | 127 | 29 | | | | | |
| | Negative | 75 | 108 | | | | | |
| | | **Composite reference** | | | | | | |
| | **Microscopy** | Positive | Negative | 61.9 (54.8–68.6) | 96.4 (91.7–98.8) | 0.537 | <0.001 | Moderate |
| | Positive | 125 | 5 | | | | | |
| | Negative | 77 | 132 | | | | | |
| Study B1 (N = 798) | | **Microscopy** | | | | | | |
| | **Schistoscope** | Positive | Negative | 93.2 (89.7–95.7) | 82.1 (78.4–85.4) | 0,723 | <0.001 | Substantial |
| | Positive | 286 | 88 | | | | | |
| | Negative | 21 | 403 | | | | | |
| Study B2 (N = 349) | | **Microscopy** | | | | | | |
| | **Schistoscope** | Positive | Negative | 96.3 (92.6–98.5) | 86.8 (80.5–91.6) | 0.837 | <0.001 | Almost perfect |
| | Positive | 183 | 21 | | | | | |
| | Negative | 7 | 138 | | | | | |
| | | **Composite reference** | | | | | | |
| | **Schistoscope** | Positive | Negative | 78.0 (72.3–83.0) | 90.9 (83.4–95.8) | 0.604 | <0.001 | Moderate |
| | Positive | 195 | 9 | | | | | |
| | Negative | 55 | 90 | | | | | |
| | | **Composite reference** | | | | | | |
| | **Microscopy** | Positive | Negative | 75.2 (69.4–80.4) | 98.0 (93.0–99.8) | 0.619 | <0.001 | Substantial |
| | Positive | 188 | 2 | | | | | |
| | Negative | 62 | 97 | | | | | |

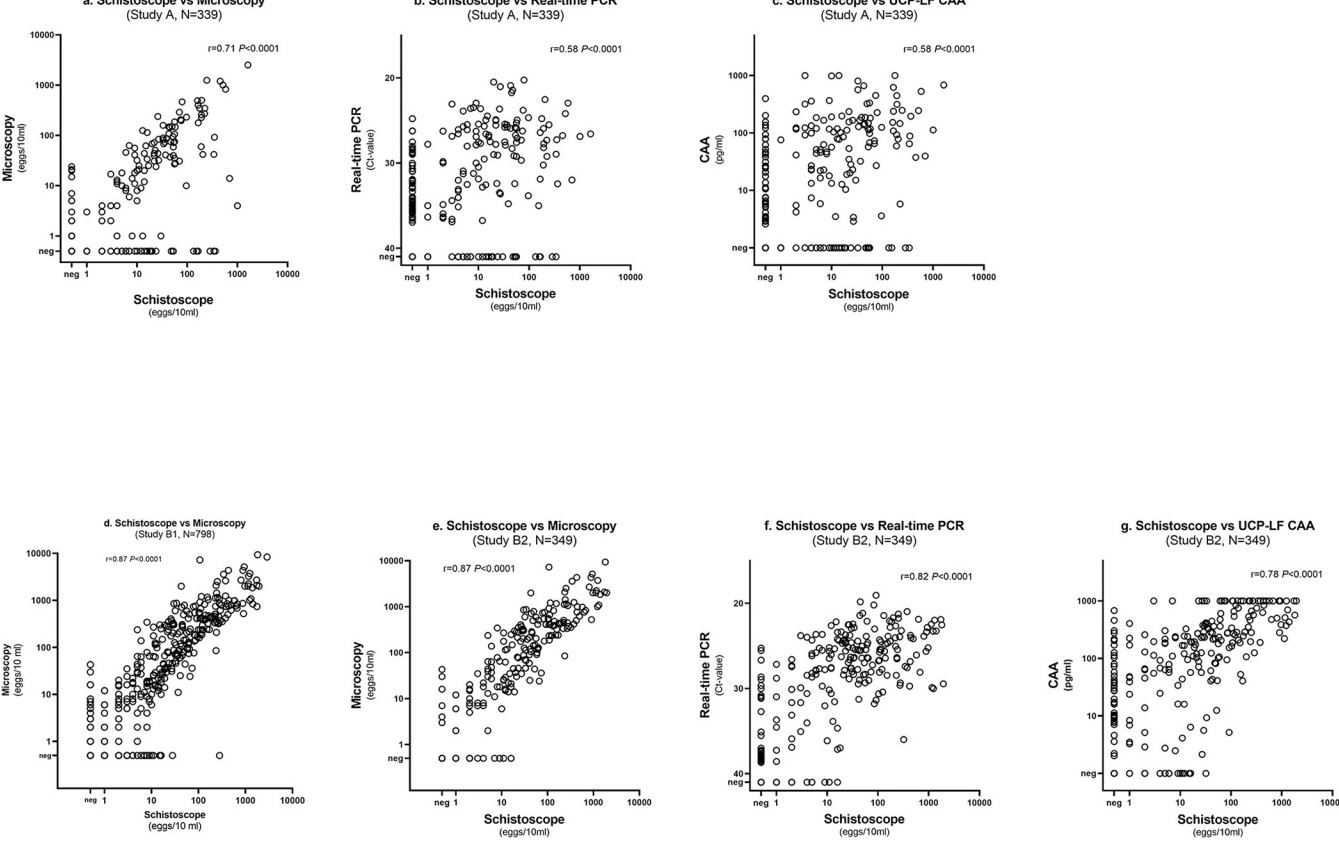

**Fig 3.** Correlation between *S. haematobium* egg counts measured by the Schistoscope and *S. haematobium* egg counts measured by conventional microscopy (a, d, e), Ct-values determined by real-time PCR (b, f) and urine CAA concentration measured by UCP-LF CAA (c, g) in study A and B.

A strong correlation was observed between egg counts estimated by the Schistoscope and conventional microscopy (r = 0.71, *P*<0.0001) (Fig 3A). A moderate correlation was observed between the Schistoscope egg counts and real-time PCR Ct-value (r = -0.58, *P*<0.0001), and CAA concentration (r = 0.58, *P*<0.0001) (Fig 3B and 3C, respectively). Bland-Altman analysis revealed that the Schistoscope tended to underscore egg counts compared to conventional microscopy, but approximately 95% of the difference in the egg count estimates between both methods fell within the limit of agreement (Fig 4A).

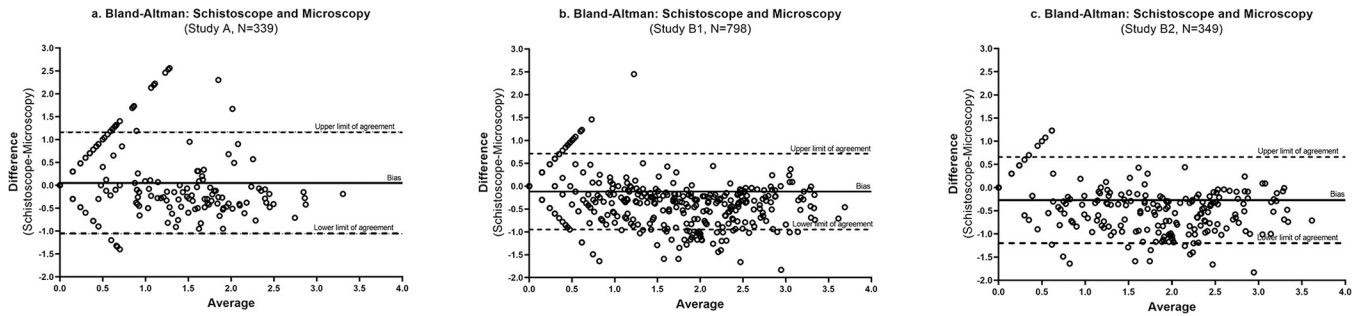

**Fig 4.** Bland-Altman analysis demonstrating the quantitative agreement between the Schistoscope and conventional microscopy in study A (a) and B (b, c).

### 3.2. Study B: Diagnostic performance of the Schistoscope on banked samples

A total of 798 samples, for which both Schistoscope and conventional microscopy results were available, were included in the analysis (Study B1). Quality control of the biobank revealed no significant difference in microscopy egg count before and after storage which confirmed the integrity of the biobank. The percentage of positive cases detected by the Schistoscope (46.9%) was higher than by conventional microscopy (38.5%). The proportion of positives with an egg count of ≥50 eggs/10 mL was substantially lower by the Schistoscope (32.6%) than by conventional microscopy (59.3%) (S1B and S1C Fig).

A subset of 349 samples had test results available from all four diagnostic tests and were further analysed (Study B2). Based on real-time PCR and UCP-LF CAA a high percentage positive was observed, 62.2% and 64.5%, respectively. The percentage of positive cases detected by the Schistoscope and conventional microscopy were similar, 58.5% and 54.4% respectively, with a significantly different median egg count (Table 1). All samples with high infection intensity were detected by the Schistoscope (S1C Fig). In addition, the Schistoscope detected 76.5% and 93.0% of samples with microscopy egg count 1–9 eggs/10mL and 10–49 eggs/10mL respectively. On the contrary, the Schistoscope found 21 additional cases with low infection intensity not detected by conventional microscopy. Of the 21 cases, 57.6% and 38.1% were confirmed by real-time PCR and UCP-LF CAA assay respectively.

A substantial to almost perfect qualitative agreement between the Schistoscope and conventional microscopy was observed in study B1 and B2 respectively (Table 2). The agreement between the Schistoscope and the CRS was similar to the agreement between conventional microscopy and the CRS. The sensitivities and specificities of the Schistoscope in study B1 and B2 when conventional microscopy was used as a reference were comparable. Furthermore, a comparable sensitivity and specificity between the Schistoscope and conventional microscopy was observed when evaluating both methods against the CRS.

A very strong correlation between the egg counts of the Schistoscope and conventional microscopy was observed in study B1 (r = 0.87; $P<0.0001$, Fig 3D) and study B2 (r = 0.93, $P<0.0001$, Fig 3E). In study B2, a lower though significant correlation was observed between the Schistoscope egg counts and PCR Ct-values (r = -0.82, $P<0.0001$) and CAA concentration (r = 0.78, $P<0.0001$) (Fig 3F and 3G). Bland-Altman analysis further demonstrated a strong quantitative agreement between the Schistoscope and conventional microscopy in both study B1 and B2 (Fig 4B and 4C) with a trend in the Schistoscope underestimating egg count.

## 4. Discussion

For the first time, we demonstrate the sensitivity and specificity of the Schistoscope with an onboard integrated AI on fresh (study A) and stored (study B) sample sets in comparison to conventional microscopy as well as to a more sensitive CRS consisting of real-time PCR and UCP-LF CAA. Five Schistoscopes were successfully transported and implemented in the parasitology laboratory of CERMEL, a reference laboratory setting within a rural part of Gabon, which is a region endemic for *S. haematobium*. All other diagnostic tests were also performed at CERMEL. Overall, the performance of the Schistoscope was non-inferior to conventional microscopy with a comparable sensitivity and a slightly lower specificity. The Schistoscope is a promising tool for urogenital schistosomiasis screening in endemic settings and offers the advantage of data connectivity and the possibility of task shifting [27–29].

Qualitatively, a moderate to almost perfect agreement between the Schistoscope and conventional microscopy was found while a fair to moderate agreement was observed when compared to the CRS. This lower agreement can mainly be attributed to the fact that the two

additional diagnostic tests included in the CRS (PCR and CAA) are more accurate, especially at low infection intensities, and these tend to be missed by the Schistoscope and/or conventional microscopy. The sensitivity of the Schistoscope was found to be non-inferior to conventional microscopy in both study A and B2. The specificity of the Schistoscope was however inferior to conventional microscopy in study A, but comparable in study B2. This is thought to be a consequence of the presence of relatively more artifacts in the freshly prepared slides (study A) compared to stored slides (study B), which the AI algorithm could not differentiate from eggs. Secondly, although samples in study A and B were obtained from the same geographical area in Gabon (Lambaréné and its surrounding villages), they were collected at different time points (~2 years apart) as well as from different populations, i.e. community-based in study A versus specific populations including pregnant women in study B. Differences in urine composition due to differential seasonal concomitant bacterial infections was assumed to explain increase in egg-like crystals formation in urine that interfered with AI detection. Manual re-analysis of the images of a selection of samples that were positive by the Schistoscope but negative by conventional microscopy, revealed that indeed crystals were present in these slides, which the AI incorrectly identified as eggs (S3 Fig).

Although the sensitivity of the Schistoscope in study A (83.1%) was comparable to previously reported results from a field setting in Nigeria (87.3%) [12], the observed specificity was significantly higher (77.0% compared to 48.9% in Nigeria) as well as the correlation between egg counts by the Schistoscope and conventional microscopy, indicating the successful re-designing and re-training of the AI algorithm [21]. The slightly lower correlation observed between the Schistoscope egg count and real-time PCR Ct-values or CAA concentration could be because of the differences in diagnostic target; eggs, egg-DNA and circulating antigen, respectively. The correlation between conventional microscopy and real-time PCR or UCP-LF CAA resulted in a similar observation (S2 Fig). The correlation observed between egg counts by conventional microscopy and Ct-values is comparable to previous findings [30]. Furthermore, although a better correlation would be expected between egg counts and Ct-values (egg-DNA), considering that they are both egg-based detection methods, it is important to note that, an egg does not have a fixed target DNA copy number. This variation is influenced by the egg developmental stage, which could account for the broad spectrum of Ct-values observed across varying infection intensities or egg count [31].

In study B overall, over 50% of the samples had only results for microscopy and the Schistoscope and rather than discarding this number of samples from our data set, it was analysed separately as B1. For both studies, all cases with high infection intensity based on conventional microscopy, known to correlate strongly with morbidity of the disease [32], were detected by the Schistoscope. Following Bland-Altman's analysis, a constant but clinically fair bias (absolute error) between Schistoscope and conventional microscopy egg counts was observed in both the fresh and stored sample sets, suggesting that the Schistoscope is slightly underestimating egg counts at a constant rate. This could be explained by the fact that with increasing infection intensity, eggs tend to overlap which could not be accurately counted by the AI algorithm (Fig 2C), as also observed previously [12]. Furthermore, the AI algorithm was designed and optimised for specificity at the expense of sensitivity, i.e. it was programmed to refrain from detecting truncated eggs located on boundaries of images so as to reduce the chances of detecting artifacts as well as eggs with lower morphological attributes. Such errors can be corrected by further optimisation of the AI algorithm in order to quantify the number of eggs more accurately. Also, in both the fresh and stored sample set, the majority of missed cases by the Schistoscope had a very low intensity of infection based on conventional microscopy ($\leq 5$ eggs/10mL), highlighting another area of focus for the next generation of the Schistoscope.

Our results indicated a better sensitivity of the Schistoscope on banked sample slides compared to fresh samples. This could be due to the difference in the infection intensity observed in the two studies. The median egg count, based on conventional microscopy, was lower in the fresh samples compared to the banked samples, which implies that the Schistoscope performs better on samples with a higher infection intensity, as also previously reported [12, 21]. Nevertheless, our results demonstrate a good performance of the Schistoscope on banked sample slides, indicating the possibility for retrospective analysis of banked sample slides in settings lacking direct access to microscopists.

In study A, the sensitivity of conventional microscopy estimated based on the CRS (62%) was lower than the sensitivity (80%) assumed for power calculations, in contrast to study B where the sensitivity (75.2%) observed was comparable. Retrospectively, the sample size calculation was limited in that it did not take into account the proportion of high-intensity infections but only incorporated prevalence, which could have had a significant impact on the sensitivity of conventional microscopy as it is known that the sensitivity of microscopy is limited in case of low intensity infections [33]. Overall, the proportion of high-intensity infections in study A was significantly lower compared to study B, resulting in a lower sensitivity of conventional microscopy as observed in study A. Despite the difference between the assumed and obtained sensitivity of conventional microscopy, we still believe our study had sufficient power to accurately determine the performance of the Schistoscope. So far, the performance of the Schistoscope has been evaluated in two endemic settings in urine samples producing promising outcomes. There is need for more performance evaluation in diverse schistosomiasis endemic settings (in urine and stool) with different climatic conditions, such as the northern part of Sahel region. Also, a cost effective analysis should be performed to support the integration of such a tool in large scale control programmes.

Limitations of this study include the time it took to analyse a slide by the Schistoscope, which on average was ~25 mins. for samples with egg counts ≥200egg/10mL even more time was needed. Furthermore, in this study a filter membrane of diameter 25mm was used (following the standard protocol of CERMEL), which also increased the time of analysis by 3-fold compared to the use of a 13mm filter membrane [12]. If a smaller filter membrane is used and the Schistoscope is programmed to stop counting when reaching 50 eggs/10mL–as this is classified as a high infection intensity and in such cases a detailed egg count is often not required [34]–the total reading time could be reduced to less than 10 mins. A tool as such would complement the existing POC-CCA urine test, which has been recommended by the WHO for *S. mansoni* infections, in settings co-endemic for *S. haematobium*. Although the Schistoscope has been fully automated, the aesthetics are currently unsatisfactory [20]. Furthermore, there is need to make the Schistoscope field-friendly and compatible to very rural settings, including the addition of a power source, improving the user interface and making it more compact and portable.

To conclude, in this study a follow-up assessment of the Schistoscope was conducted in a rural laboratory setting in Gabon, further validating its potential as a digital diagnostic tool for the identification and quantification of *S. haematobium* eggs in freshly collected as well as banked urine sample slides. Although the specificity of the Schistoscope could still be improved, its overall performance was non-inferior to conventional microscopy hence, a promising tool for urogenital schistosomiasis screening in endemic settings.

## Supporting information

**S1 Checklist. STARD-2015-Checklist.**
(DOCX)

**S1 Fig.** Agreement between the Schistoscope and microscopy per category of infection intensity in study A and B.
(TIF)

**S2 Fig.** Correlation between *S. haematobium* egg counts measured by the conventional microscopy and Ct-values determined by by real-time PCR (a, c) and urine CAA concentration.
(TIF)

**S3 Fig. Image showing crystal incorrectly detected as an egg by the Schistoscope.**
(TIF)

**S1 Manual. Schistoscope user manual.**
(PDF)

**S1 Video. Video showing the Schistoscopes running in the laboratory.**
(MOV)

**S1 Raw Dataset.** Overall raw dataset containing data for Schistoscope validation on fresh urine samples (study A), Banked slides (study B) and quality control of banked slides.
(XLSX)

## Acknowledgments

We extend our appreciation to Mermoz Ndong-Essone Ondong, Danny Carrel Manfoumbi Mabicka, Moutsinga Dalia Coralline ep Lehoumbou, Elsy Myrna N'noh Dansou, Marguerite Nzame Ngome, and the coordination team, along with all the members of Immuno-Epi research group of CERMEL. We thank Bertrand Lell for IT support in the field. Our gratitude also goes to Jean-Aimé Massande Ndzokou's and the CERMEL field team for their valuable contributions to the advancement of this project.

## Author Contributions

**Conceptualization:** Brice Meulah, Pytsje T. Hoekstra, Michel Bengtson, Cornelis Hokke, Ayola Akim Adegnika, Lisette van Lieshout.

**Data curation:** Brice Meulah, Prosper Oyibo, Pytsje T. Hoekstra.

**Formal analysis:** Brice Meulah, Pytsje T. Hoekstra.

**Funding acquisition:** Cornelis Hokke, Temitope Agbana, Jan Carel Diehl, Ayola Akim Adegnika, Lisette van Lieshout.

**Investigation:** Brice Meulah, Prosper Oyibo, Paul Alvyn Nguema Moure, Moustapha Nzamba Maloum, Romeo Aime Laclong-Lontchi, Yabo Josiane Honkpehedji, Michel Bengtson, Paul L. A. M. Corstjens.

**Methodology:** Brice Meulah, Pytsje T. Hoekstra, Michel Bengtson, Lisette van Lieshout.

**Project administration:** Pytsje T. Hoekstra, Michel Bengtson, Lisette van Lieshout.

**Resources:** Ayola Akim Adegnika.

**Software:** Brice Meulah, Prosper Oyibo.

**Supervision:** Pytsje T. Hoekstra, Michel Bengtson, Cornelis Hokke, Temitope Agbana, Jan Carel Diehl, Ayola Akim Adegnika, Lisette van Lieshout.

**Validation:** Pytsje T. Hoekstra, Lisette van Lieshout.

**Visualization:** Brice Meulah.

**Writing – original draft:** Brice Meulah.

**Writing – review & editing:** Brice Meulah, Prosper Oyibo, Pytsje T. Hoekstra, Moustapha Nzamba Maloum, Yabo Josiane Honkpehedji, Michel Bengtson, Cornelis Hokke, Paul L. A. M. Corstjens, Temitope Agbana, Jan Carel Diehl, Ayola Akim Adegnika, Lisette van Lieshout.

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
