## [Decision Letter · Decision Letter 0]

2 Jan 2024

Dear Mr Meulah,

Thank you very much for submitting your manuscript "Extended laboratory validation of the performance of an artificial intelligence-based digital microscope (Schistoscope) in Lambaréné (Gabon) for automated detection of Schistosoma haematobium eggs in urine" for consideration at PLOS Neglected Tropical Diseases. As with all papers reviewed by the journal, your manuscript was reviewed by members of the editorial board and by several independent reviewers. In light of the reviews (below this email), we would like to invite the resubmission of a significantly-revised version that takes into account the reviewers' comments. 

In this study Meulah and colleagues optimise a quantitative methodology to accurately diagnose schistosomiasis. Even though the study is timely and within the scope of the journal, several issues raised by the three reviewers and the editor need first to be thoroughly addressed before considering it suitable for publication. 

• The Title is probably too long, it could be shorter and punchier. Please consider something along these lines:

“Artificial intelligence-based digital microscopy for automated detection of Schistosoma haematobium eggs in urine.” 

• Line 139: “The Schistoscope was assumed to have a sensitivity and specificity non-inferior to conventional microscopy, which were realistically assumed to be 80% and 90%, respectively based on field expert estimates using real-time PCR and UCP-LF CAA.” Provide a reference to back up the sensitivity and specificity of conventional microscopy.

• Line 247: Please, provide reference to support the selection criteria for levels of infection intensity. 

• Results headings – consider writing headings that are more descriptive and highlight the main/ key findings described in the section

• PCR; is the positive control the same as the internal control? What kind of normalization is used during the qPCR analysis (ratio unknown: control ? What were the Cq values of the NTC? Did the authors run a standard curve along with controls?). All these technical points are extremely important not only because this study focuses on describing a new diagnostic methodology, but for the sake of transparency and reproducibility.

We cannot make any decision about publication until we have seen the revised manuscript and your response to the reviewers' comments. Your revised manuscript is also likely to be sent to reviewers for further evaluation.

Sincerely,

Gabriel Rinaldi, M.D., Ph.D.

Academic Editor

Aaron Jex

Section Editor

In this study Meulah and colleagues optimise a quantitative methodology to accurately diagnose schistosomiasis. Even though the study is timely and within the scope of the journal, several issues raised by the three reviewers and the editor need first to be thoroughly addressed before considering it suitable for publication. 

• The Title is probably too long, it could be shorter and punchier. Please consider something along these lines:

“Artificial intelligence-based digital microscopy for automated detection of Schistosoma haematobium eggs in urine.” 

• Line 139: “The Schistoscope was assumed to have a sensitivity and specificity non-inferior to conventional microscopy, which were realistically assumed to be 80% and 90%, respectively based on field expert estimates using real-time PCR and UCP-LF CAA.” Provide a reference to back up the sensitivity and specificity of conventional microscopy.

• Line 247: Please, provide reference to support the selection criteria for levels of infection intensity. 

• Results headings – consider writing headings that are more descriptive and highlight the main/ key findings described in the section

• PCR; is the positive control the same as the internal control? What kind of normalization is used during the qPCR analysis (ratio unknown: control ? What were the Cq values of the NTC? Did the authors run a standard curve along with controls?). All these technical points are extremely important not only because this study focuses on describing a new diagnostic methodology, but for the sake of transparency and reproducibility.

Reviewer's Responses to Questions

**Key Review Criteria Required for Acceptance?**

**Methods**

-Are the objectives of the study clearly articulated with a clear testable hypothesis stated?

-Is the study design appropriate to address the stated objectives?

-Is the population clearly described and appropriate for the hypothesis being tested?

-Is the sample size sufficient to ensure adequate power to address the hypothesis being tested?

-Were correct statistical analysis used to support conclusions?

-Are there concerns about ethical or regulatory requirements being met?

Reviewer #1: Yes

Reviewer #2: Minor additions of detail are needed 

• The legend for figure 1 could be more descriptive

• Line 152 – you say Whatman membranes were used to capture the eggs but how was this done e.g. using a syringe and filter holder ? 

• Line 213-224 – some more detail is needed 

o What is DEC 

o What was the internal control amounts used (DNA, primers and probe)

o What was the Schisto positive control 

o Why was 40 cycles used as the cut off 

o What was the CT value of the internal control 

o How has the internal control been validated as a multiplex with the ITS primers? 

• Line 243 – what is CRS ? write in full before using the acronym

Reviewer #3: The objectives of the study are clearly articulated with a clear testable hypothesis, and correct statistical analysis were used to support conclusions stated. However, the study design is not quite clear. To me, it is not necessary to have the B1 and B2 components; Just having fresh samples and banked samples should be enough to avoid unnecessary complexity. The B1 component can be deleted. 

Also, the sample size calculation seems not appropriate, it seems like the formula for a cross sectional survey was used for sample size calculations. I didn’t attempt to calculate the sample size but I do think that a different formula integrating the minimal difference expected between both tests (Schistoscope vs microscopy and Schistoscope vs DNA- or Antigen-based assays) should be used (please see Page 10, Lines 245-247 for the hypothesis). 

Regarding the ethical regulatory requirements (please see Page 7, Lines 135-137), this is not a clinical trial, and I am wondering why this study was registered on clinicaltrials.gov. If this study just took advantage of an existing study, it is worth to mention it. Also, the authors declared that they provided treatments to all infected individuals. Knowing that the test and treat procedures are sometimes complicated, especially when the test cannot be completed in a few minutes, I am curious to know how the authors managed to find and treat all those positive patients. This can be helpful for other studies.

**Results**

-Does the analysis presented match the analysis plan?

-Are the results clearly and completely presented?

-Are the figures (Tables, Images) of sufficient quality for clarity?

Reviewer #1: Yes

Reviewer #2: A minor addition to the results would be beneficial

• Did you find other discrepancies that are challenging to interpret – eg egg positive but CAA neg and DNA neg.

Reviewer #3: The results are clearly presented and match the analysis plan. However, one of the main assets of this technology/device is the ability to be used for the diagnosis of schistosomiasis on banked samples. However, the design developed by the authors do not fully enable to better capture that asset. Indeed, it would have been interesting to have the results of conventional microscopy for all the samples, not only for a subset of samples used as quality control. Even if quality control should be considered, I am quite curious to have the results of the quality control to be able to figure out to what extent was the variation between the two readings, and whether it was unidirectional or random. This should be clearly presented and discussed.

Also, the issue of artefact raised at page 20, Lines 350-358 can be important for conventional microscopy as well. As such, it would have been worth to analyze all the banked slides, or present and discuss the results of quality control.

Page 9, Lines 191-193. It is unclear how the discrepancies between conventional microscopy readings between the two readers was done? Was a third reader involved as adjudicator or a third reading was done by the same two microscopists? This should be stated clearly.

**Conclusions**

-Are the conclusions supported by the data presented?

-Are the limitations of analysis clearly described?

-Do the authors discuss how these data can be helpful to advance our understanding of the topic under study?

-Is public health relevance addressed?

Reviewer #1: Yes

Reviewer #2: The conclusions are well supported and the limitations are clearly described. I think the thoughts around the egg number cutoff to categorise intensity rather than actual egg counts is a good way forward. 

 other minor comment are:

• Line 381 – could the difference also relate to the fact you did not bead beat the eggs so the DNA may not have been released ? 

• Mention the potential for screening stool?

Reviewer #3: The conclusions are globally supported by the data presented. However, there are a number of issues that need to be addressed: 

First and foremost, the main conclusion or finding of the manuscript is not convincing enough. In fact, the authors found that Schistoscope is non inferior to conventional microscopy which is already a poor diagnostic test especially when it comes to talk about transmission interruption or elimination. It is well known and widely accepted that when the parasitic loads are low, the sensitivity of microscopy is low and more sensitive tools are therefore needed. This should be raised as a limitation of the study/manuscript. I would like to acknowledge that Schistoscope has a good potential, and the tool can still be improved. Indeed, I had the opportunity to review the manuscript presenting the early versions of the device which was clearly inferior to microscopy. The current version of the device, even if non-inferior to conventional microscopy, is significantly less performant than DNA- and antigen-based assays (please see figures at page 16), and it might be worth to clearly indicate the potential applications of Schistoscope.

It is unclear to me how the authors try to justify the difference between Schistoscope and PCR or CAA (Page 20, Lines 372-375). There is not just a matter of target but sensitivity and specificity of the techniques. This should be clearly explained.

The authors mentioned the limitations (time of Schistoscope operation) of the study (Page 22, Lines 420-433) that raised another concern or issue. It would have been worth to compare the time of operations of Schistoscope to that of conventional microscopy in the same circumstances. This is important and aligns with the applications of Schistoscope mentioned above.

**Editorial and Data Presentation Modifications?**

Reviewer #1: Minor Revision

Reviewer #2: (No Response)

Reviewer #3: This manuscript needs some language editing or correction of typos. For example, please consider “as previously described” not “described previously” (page 9 line 209); “internal transcribed spacer” not “internally transcribed spacer” (page 9 line 210); “ct” not “cq” (page 9 line 233) …

Authors should avoid some of abbreviations or explain them at their first used. For example “UCP-LF-CAA” and “CRS” in the abstract … 

Some of the references used in the manuscript are not appropriately presented. For example, WHO should be numbered as all other references …

**Summary and General Comments**

Reviewer #1: See attached file

Reviewer #2: This study fully evaluates the sensitivity and specificity of the Schisoscope for eggs detection in urine. This is a large and detailed study that warrants publication. The methods are sound and the data interpretation has been well executed. I look forward to further developments of the Schistoscope as its limiations are addressed.

Reviewer #3: The manuscript submitted by Meulah and colleagues entitled “Extended laboratory validation of the performance of an artificial intelligence-based digital microscope (Schistoscope) in Lambaréné (Gabon) for automated detection of Schistosoma haematobium eggs in urine” aimed to validate the performance of a novel AI-based diagnostic tool, the Schistoscope, for the detection and quantification of Schistosoma haematobium eggs in urine using conventional microscopy and a composite diagnosis (real-time PCR and a lateral flow assay) as a reference. Although the accurate diagnosis of schistosomiasis is of high interest and an up-to-date issue in the framework of transmission interruption/elimination (in its 2021-2030 roadmap, one of the critical actions recommended by the WHO is the development of diagnostic tests, including standardized point-of-care diagnostic), the paper lacks clarity, especially in its design.

PLOS authors have the option to publish the peer review history of their article (what does this mean?). If published, this will include your full peer review and any attached files.

Reviewer #1: Yes: Jean T. Coulibaly

Reviewer #2: No

Reviewer #3: No
---

## [Editor Report · Decision Letter 1]

5 Feb 2024

Dear Mr Meulah,

We are pleased to inform you that your manuscript 'Validation of artificial intelligence-based digital microscopy for automated detection of Schistosoma haematobium eggs in urine in Gabon' has been provisionally accepted for publication in PLOS Neglected Tropical Diseases.

Best regards,

Gabriel Rinaldi, M.D., Ph.D.

Academic Editor

Aaron Jex

Section Editor

---

## [Editor Report · Acceptance letter]

19 Feb 2024

Dear Mr Meulah,

We are delighted to inform you that your manuscript, "Validation of artificial intelligence-based digital microscopy for automated detection of Schistosoma haematobium eggs in urine in Gabon," has been formally accepted for publication in PLOS Neglected Tropical Diseases.

Best regards,

Shaden Kamhawi

co-Editor-in-Chief

Paul Brindley

co-Editor-in-Chief
